# Livelihood and Food Security in the Context of Sustainable Agriculture: Evidence from Tea Agricultural Heritage Systems in China

**DOI:** 10.3390/foods13142238

**Published:** 2024-07-16

**Authors:** Jilong Liu, Chen Qian, Xiande Li

**Affiliations:** Institute of Agricultural Economics and Development, Chinese Academy of Agricultural Sciences, Beijing 100081, China; ljlong95@163.com (J.L.); qianchen@caas.cn (C.Q.)

**Keywords:** sustainable agriculture, agricultural heritage systems, livelihood security, food security, Anxi Tieguanyin Tea Culture System, Fuding White Tea Culture System

## Abstract

The conservation of agricultural heritage systems (AHSs) has played a pivotal role in fostering the sustainable development of agriculture and safeguarding farmers’ livelihoods and food security worldwide. This significance is particularly evident in the case of tea AHSs, due to the economic and nutritional value of tea products. Taking the Anxi Tieguanyin Tea Culture System (ATTCS) and Fuding White Tea Culture System (FWTCS) in Fujian Province as examples, this study uses statistical analyses and a multinomial logistic regression model to assess and compare farmer livelihood and food security at the tea AHS sites. The main findings are as follows. First, as the tea industries are at different stages of development, compared with agricultural and non-agricultural part-time households, the welfare level of pure agricultural households is lowest in the ATTCS, while welfare is the highest in the FWTCS. Second, factors such as the area of tea gardens and the number of laborers significantly affect farmers’ livelihood strategies transformation from pure agricultural households to agricultural part-time households in the ATTCS and FWTCS. Third, the high commodity rate of tea products, combined with compound cultivation in tea gardens, provides local people with essential sources of income, food, and nutrients, so as to improve food security in the ATTCS and FWTCS. These findings are essential for designing policies to ensure farmers’ livelihoods and food security through AHSs and other sustainable agriculture.

## 1. Introduction

Sustainable agriculture is widely defined as the efficient production of safe, high-quality agricultural products, in a way that protects the natural environment, improves the social-economic conditions of farmers and local communities, and safeguards the health and welfare of all farmed species [1]. Characterized by social, economic, and environmental sustainability, sustainable agriculture has been instrumental in ensuring food and nutrition security, which aligns with Sustainable Development Goal 2 [2,3]. It is also a key point for achieving other Sustainable Development Goals (SDGs) [4]. In practice, sustainable agriculture includes precision agriculture (e.g., crop-dusting), conservation agriculture (e.g., zero-tillage), and ecological agriculture (e.g., intercropping) [5,6,7]. The adoption of these practices has significantly contributed to local food, nutrition, and livelihood security [8,9].

However, with the rapid pace of industrialization and urbanization, many traditional agricultural practices are being gradually abandoned, resulting in the emergence of increasingly unsustainable agricultural systems [10,11]. To address this challenge, international organizations and local governments have jointly exerted efforts to recognize agricultural heritage systems (AHSs) as a means to preserve traditional agricultural wisdom and promote the sustainable development of agriculture. One such initiative is the project of the Food and Agriculture Organization (FAO) of the United Nations (UN) on the dynamic conservation and adaptive management of Globally Important Agricultural Heritage Systems (GIAHSs), which was established in 2002. As of May 2024, there were 86 GIAHSs recognized [12]. Drawing inspiration from the FAO’s GIAHS project, China has embarked on its own identification and conservation of Nationally Important Agricultural Heritage Systems (China-NIAHSs) since 2012, with 188 China-NIAHSs as of May 2024 [13,14]. In recent years, the significance of AHS conservation in sustainable agriculture has gained worldwide recognition. AHSs not only maintain agricultural biodiversity and form a wealth of indigenous knowledge, but more importantly, they also provide various products and services that ensure local livelihoods and food security, ultimately improving people’s quality of life [15,16,17].

Tea AHSs serve as a prime example for how AHSs contribute to local livelihood and food security given their significant economic, ecological, and social value. The cornerstone of tea AHSs lies in four key elements: tea germplasm resources, ecological tea gardens, tea-making skills, and tea cultural activities. Firstly, the unique and abundant tea germplasm resources provide vital biological assets for the promotion of distinctive agriculture and organic farming, enhancing the value of agricultural products, and creating comparative advantages in agriculture, thereby contributing to economic development and income growth [18]. Secondly, the construction of ecological tea gardens is conducive to the certification of tea products. Certified tea products tend to be more marketable due to their higher prices and increased brand value, thus increasing the income of tea farmers [19,20]. Thirdly, the organization of tea-making competitions and the management of tea associations enhances the bond between farmers’ tea production and the accumulation of their cultural capital. This connection is vital for maintaining and promoting the social stability and economic development of the community [21].

However, discussions regarding the contribution of AHSs, particularly tea AHSs, to livelihood and food security remain inadequate. In existing studies, livelihood security primarily encompasses the livelihood capital and economic outcomes of those residing in AHS sites, while food security mainly refers to the products provided by the AHSs [22,23]. Moreover, a significant limitation is that most studies are individual case studies, which makes it difficult to generalize their findings to all AHS sites [24]. In addition, the assessment of farmer welfare, which is crucial for livelihood security, is generally overlooked [25]. To address these gaps, this study assesses farmer livelihood and food security in tea AHS sites, specifically the Anxi Tieguanyin Tea Culture System (ATTCS) and Fuding White Tea Culture System (FWTCS) in Fujian Province, China. This study offers a twofold contribution to the literature. On the one hand, it proposes a conceptual framework discussing the role and contribution of AHSs to the development and promotion of sustainable agriculture. On the other hand, it enriches the existing research on farmer livelihood and food security by conducting a comparative analysis across various tea AHS sites.

The remainder of this paper is organized as follows. Section 2 reviews the literature on the relationships between AHSs, sustainable agriculture, and livelihood and food security. Section 3 describes the study area, data sources, and empirical methods. Section 4 analyzes the results. Section 5 discusses this study’s findings. Finally, Section 6 summarizes this study’s contributions and provides policy suggestions.

## 2. Literature Review

### 2.1. AHSs and Sustainable Agriculture

Sustainable agriculture (SA) encompasses various definitions from different perspectives. According to the FAO, SA involves “adopting fundamental approaches to the utilization and conservation of natural resources, along with technological and institutional innovations, to ensure that the demand of present and future generations for agricultural products are continuously met” [26]. SA includes advancements in agricultural management practices and technologies, such as precision agriculture, conservation agriculture, and ecological agriculture. The main goals of SA are to improve environmental health, economic profitability, and social and economic equity [27]. As the significance of SA in food security grows, its definition increasingly emphasizes environmental, economic, and social sustainability [2]. Environmental sustainability involves natural-based practices that protect the environment, improve soil quality, and enhance biodiversity by applying organic fertilizers, botanical pesticides, and natural predators [28]. Economic sustainability refers to using green production technologies that increase agricultural yield and production capacity, thereby improving economic benefits [29]. Social sustainability contributes to societal stability through sustainable natural resource management, such as traditional practices for the rational utilization of water resources [30].

AHSs, defined as remarkable land-use systems and landscapes that are rich in globally significant biological diversity evolving from the co-adaptation of a community with its environment and its needs and aspirations for sustainable development, are a prominent example and demonstration of SA, particularly in remote areas and among indigenous communities. These systems are evaluated based on five selection criteria: food and livelihood security, agrobiodiversity, local and traditional knowledge systems, cultures, value systems and social organizations, and landscape and seascape features [31]. AHSs emphasize traditional agricultural production systems with a profound history, centered around rich landscapes, environmentally friendly agricultural production practices, and the inheritance of indigenous cultures [32,33]. As typical agro-ecological systems, one of the main characteristics of AHSs is sustainability, which is reflected in their adaptability to extreme natural conditions, maintaining the livelihood security of residents and promoting harmonious community development [34,35]. Put differently, AHSs reflect principles related to environmental, economic, and social sustainability [36]. For instance, China’s Qingtian rice-fish co-culture system, designated as a GIAHS in 2005, exemplifies environmental sustainability. Its multi-cropping and breeding approach of rice and fish fosters a mutually beneficial symbiotic relationship that minimizes the need for chemical fertilizers, pesticides, and insecticides, thus safeguarding the ecological integrity of rice fields. On the one hand, fish in the field collide with the rice stalks, causing pests to fall into the water and thus reducing pest infestations. On the other hand, they nibble on the barnyard grass to reduce weed damage [37]. The Longsheng Longji Terraces in China, recognized as a GIAHS in 2018, illustrate economic sustainability. The multifunctional development of terraces has diversified farmers’ income sources, including agricultural income, subsidies, tourism dividends, and non-agricultural business income, which encourages farmers to continue terraced production and preserve the local landscape [38]. With regard to social sustainability, Chiloé agriculture in Chile, designated as a GIAHS in 2011, presents a good example. Rich local potato varieties are protected through traditional knowledge that is passed down orally and through strong social networks among generations of farmers [39]. Hence, AHSs exemplify SA by emphasizing the importance of environmental, economic, and social sustainability within traditional agricultural systems (Figure 1). 

### 2.2. Tea AHSs and Livelihood and Food Security

Based on the list of designated AHSs, various types of AHSs exist according to their main agricultural components, including fruit, tea, rice, and animal husbandry. Notably, there are 5 tea GIAHSs and 23 tea China-NIAHSs. In tea AHSs, the tea plantation system is central, with tea culture influencing the entire production, processing, and consumption process [40]. Compared to other types of AHSs, tea culture within tea AHSs uniquely embodies agricultural tradition, reflecting local residents’ spiritual beliefs and positively impacting their lifestyles. Moreover, the abundant and diverse biological resources of tea AHSs can provide a variety of products for the residents of the heritage site. By implementing the composite cultivation of tea trees, grains, fruit trees, and herb crops, tea AHSs not only produce tea and related processed products but also diversify agricultural, forestry, fruit, and livestock outputs, meeting the food and nutritional needs of local farmers [41,42]. At tea AHS sites, tea products are the main economic source for farmers, while other agricultural products grown in tea gardens are primarily used for their own consumption, ensuring food security and nutrition [43]. Additionally, the tea industry chain has expanded to include tea planting, processing, sales, and tourism, creating numerous employment opportunities and providing a major source of livelihood for local people [44].

## 3. Materials and Methods

### 3.1. Study Area

The ATTCS is a comprehensive agricultural production system centered around the traditional Tieguanyin variety. It integrates breeding, planting, cultivation, protection, harvesting, and tea culture, fostering biodiversity, ecosystem functions, and a unique human–nature landscape. As the origin of oolong tea, it was identified as a China-NIAHS in 2014 and designated as a GIAHS site by the FAO in 2022. The ATTCS is located in Anxi County of Quanzhou City in the southeastern of Fujian Province, China, and mainly includes Lutian Town, Xiping Town, and Huqiu Town (Figure 2). The geographical location of Anxi County is 117°36′~118°17′ E, 24°50′~25°26′ N, and mountains and hills dominate its terrain. The climate of Anxi County is a subtropical maritime monsoon, with an average annual rainfall of 1782.4 mm. Due to its geographical location and favorable natural ecological conditions, Anxi County is the largest tea-producing county in China, also known as the Chinese tea capital. In 2022, the area of tea gardens in Anxi County was 43,552.60 hectares, and the per capita disposable income of rural residents of Anxi County was CNY 22,341 (equivalent to USD 3321.66). It was estimated that more than 900 thousand people worked in tea-related industries in Anxi County in 2022.

The FWTCS is another comprehensive agricultural production system centered on the traditional white tea variety. Similarly, it integrates breeding, planting, cultivation, protection, harvesting, and tea culture, also nurturing biodiversity, ecosystem functions, and a distinct human–nature landscape. The FWTCS was designated as a China-NIAHS in 2017. The FWTCS is situated in the Fuding City of Ningde City in the northeast of Fujian Province, China, and mainly includes Guanyang Town, Diantou Town, Bailin Town, Panxi Town, and Taimushan Town (Figure 2). The geographical location of Fuding City is 119°55′~120°43′ E, 26°52′~27°26′ N, and mountains and hills dominate its terrain. The climate of Fuding City is a subtropical maritime monsoon, with an average annual rainfall of 1730.61 mm. Owing to its geographical location and favorable natural ecological conditions, Fuding City is the world-famous origin of white tea. It is also the largest producer and exporter of white tea in China. In 2022, the area of Fuding City’s tea gardens was 19,382.93 hectares, and the per capita disposable income of rural residents in Fuding City was CNY 23,198 (equivalent to USD 3449.08). It was estimated that more than 380 thousand people worked in tea-related industries in Fuding City in 2022.

### 3.2. Data Sources

In November 2019 and July 2020, depending on the size of the tea garden area and the characteristics of tea cultural resources, we selected three villages in Lutian Town, eight in Xiping Town, and three in Huqiu Town in the ATTCS as the surveyed administrative villages. Then, 10–20 rural households in each village were randomly selected for interviews and questionnaire surveys. Similarly, in March and July 2023, 16 administrative villages in Guanyang Town, Diantou Town, Bailin Town, Panxi Town, and Taimushan Town in the FWTCS were selected for investigation. The number of valid questionnaires was 214 and 217 for the ATTCS and FWTCS, respectively (Table 1). The main contents of the questionnaire included farmers’ basic information, households’ agricultural production and management situations, and households’ livelihood capital, strategies, income, and welfare.

Since men are the main decision-makers in agricultural production and have a better understanding of the input–output situation of agricultural planting in rural households in both the ATTCS and FWTCS, most respondents were male, accounting for 92.06% and 84.79% of respondents, respectively. The most common age group among respondents was 46–65 years, accounting for 62.15% and 61.29%, respectively. The largest proportion of respondents had a junior high school education, accounting for 53.27% and 30.88%, respectively. Households consisting of four to six members were the most prevalent, comprising 58.41% and 64.98%, respectively. The majority of tea gardens were between 0.13 and 0.67 hectares, representing 61.68% and 67.74%, respectively (Table 2).

### 3.3. Classification of Livelihood Strategies

Household livelihood strategies are typically classified according to the proportion of agricultural income in households’ total income or based on the main livelihood activities of the household labor force [45,46]. In rural China, it is common to classify households’ livelihood strategies based on their income structures. For example, the National Bureau of Statistics of China proposed in 2005 that households’ livelihood strategies can be divided into three types based on income structure: pure agricultural, part-time agricultural, and non-agricultural households. Pure agricultural households have agricultural income accounting for more than 90% of their total income. Part-time agricultural households have agricultural income constituting 10% to 90% of their total income. Non-agricultural households have agricultural income accounting for less than 10% of their total income [47].

Farmers’ agricultural income at the AHS sites mainly comes from planting grain or cash crops, growing fruit trees and forestry, and breeding livestock and poultry. Their non-agricultural income mainly comes from working as migrant laborers and receiving government subsidies. Therefore, through an analysis of the existing research [48] and the combination of the investigation of the ATTCS and FWTCS, the types of households’ livelihood strategies were divided into four types in this study: pure agricultural households (agricultural income accounts for more than 90% of household income), agricultural part-time households (agricultural income accounts for 50–90% of the total), non-agricultural part-time households (agricultural income constituting 10% to 50% of household income), and non-agricultural households (agricultural income accounts for less than 10% of household income).

### 3.4. Livelihood Capitals Measurement

#### 3.4.1. Evaluation Indicators of Livelihood Capitals

According to the sustainable livelihood approach developed by DFID [49,50], farmer livelihood capital consists of natural, physical, financial, human, and social capital. Farmers’ livelihood capitals should include cultural capital at AHS sites due to the strong correlation between farmers’ livelihoods and local culture [51].

Natural capital generally refers to the land, water, and biological resources that farmers can use to sustain their livelihoods. Physical capital refers to farm household infrastructure and equipment related to production and living processes. Financial capital generally refers to cash deposits, available loans, borrowings, or other easily realized forms of saving. Human capital refers to the personal labor skills, learning ability, and health possessed by families or individuals in the process of engaging in productive labor. Social capital refers to the social resources that farmers can use to carry out livelihood activities, such as social networks and social organizations. Cultural capital refers to the national culture, folk customs, traditional technologies, and valuable knowledge for livelihood activities. The specific indicators of livelihood capitals are listed in Table 3.

#### 3.4.2. Data Normalization

Indicators of livelihood capital cannot be directly compared because they have different dimensions. The maximizing deviation method [52] was used to normalize the data as follows:(1)$$r_{ij}=\frac{x_{ij}-min(x_{j})}{\max\left(x_{j}\right)-min(x_{j})},$$
where *r_ij_* represents the standardized value of livelihood capitals indicator *j* for sample *i*, *x_ij_* represents the actual value of livelihood capitals indicator *j* for sample *i*, and max(*x_j_*) and min(*x_j_*) are the maximum and minimum values of livelihood capitals indicator *j*, respectively.

#### 3.4.3. Measurement of Weights

After data normalization, a zero value inevitably appears, affecting the subsequent calculation results of entropy. Hence, *r_ij_* must be shifted horizontally: *r’_ij_* = *r_ij_* + *k*. The constant *k* represents translation amplitude. The smaller the value of *k*, the closer it is to the real situation. In this study, *k* = 0.0001 was selected.

The entropy evaluation method [23,53] can calculate the importance of each indicator according to the “information” provided, which can solve the problem of weighted accounting of multiple indicators and is an essential method of objective weighting to avoid subjectivity. Generally, the entropy evaluation method includes the following calculation processes:(2)$$p_{ij}=\frac{{r^{\prime }}_{ij}}{\sum_{i=1}^{m}{r^{\prime }}_{ij}}$$
where *p_ij_* is the contribution degree of livelihood capitals indicator *j* for sample *i*.
(3)$$S_{j}=-\frac{1}{\ln m}\sum_{i=1}^{m}p_{ij}\ln p_{ij}$$
where *S_j_* is the entropy value of livelihood capitals indicator *j*.
(4)$$w_{j}=\frac{1-S_{j}}{\sum_{i=1}^{m}1-S_{j}}$$
where *w_j_* represents the weight of livelihood capitals indicator *j*.
(5)$$Y_{i}=\sum_{j=1}^{M}r_{ij}w_{j}$$

In Equation (5), *Y_i_* represents the livelihood capital value of sample *i*, and *M* is the number of types of livelihood capital. When the value of *M* is six, livelihood capital consists of natural, physical, financial, human, social, and cultural capital.

### 3.5. Analysis of the Impact of Livelihood Capitals on Livelihood Strategies

To identify the key livelihood capitals influencing the transformation of farmer livelihood strategies from pure agricultural households to agricultural and non-agricultural part-time households, we set up a multinomial logistic regression model [54] as follows:(6)$$\ln\left(p_{y2}/p_{y1}\right)=\beta_{210}+\beta_{211}X_{1}+,\cdots ,+\beta_{21m}X_{i}$$
(7)$$\ln\left(p_{y3}/p_{y1}\right)=\beta_{310}+\beta_{311}X_{1}+,\cdots ,+\beta_{31m}X_{i}$$

In Equations (6) and (7), *P_y_*_1_ = 1 for a farmer livelihood strategy that is purely agricultural, *P_y_*_2_ = 2 if part-time agricultural, and *P_y_*_3_ = 3 if part-time non-agricultural. *β*_210_, *β*_211_, …, *β*_21*m*_ and *β*_310_, *β*_311_, …, *β*_31*m*_ are the coefficients to be estimated. They represent the variation in the dependent variable caused by a variation in one unit of the corresponding independent variable. When the estimated coefficient is greater than zero, the probability of the occurrence of the dependent variable increases with an increase in the corresponding independent variable, while the other variables remain constant. In contrast, when the estimated coefficient is less than zero, the probability of the occurrence of the dependent variable decreases with an increase in the corresponding independent variable, while the other variables remain constant. *X*_1_, …, *X_i_* are independent variables.

## 4. Results

### 4.1. Livelihood Strategies and Livelihood Activities

Combined with the field investigations, households in the ATTCS and FWTCS were classified into four types according to the proportion of agricultural income in household income: pure agricultural households (HT1), agricultural part-time households (HT2), non-agricultural part-time households (HT3), and non-agricultural households (HT4). In the ATTCS, the number of HT1 and HT2 households accounted for 44.49% and 37.38%, respectively, and the total accounted for 81.87%, indicating that their livelihood strategies were still more dependent on agriculture. This result is mainly because the tea industry chain in the ATTCS is relatively long, and more farmers can engage in tea planting, processing, selling, and other links. For FWTCS households, the numbers of HT1, HT2, and HT3 households were relatively close, accounting for 32.45%, 34.57%, and 31.38%, respectively, indicating that their livelihood strategies tended to be part-time. The number of HT4 households in the ATTCS and FWTCS were the lowest (Figure 3). Therefore, considering the small sample size of HT4 households, the subsequent analysis primarily focuses on the livelihoods of HT1, HT2, and HT3 households.

Furthermore, the livelihood activities of households in the ATTCS were more diversified than those in the FWTCS. In the ATTCS, the main livelihood activities of HT1 households were tea planting and livestock and poultry breeding, whereas the main livelihood activities of HT2 and HT3 households were tea planting, non-agricultural employment, and livestock and poultry breeding. The scale of tea planting for HT1 households was generally larger than that of HT2 and HT3 households in the ATTCS. For FWTCS households, however, the main livelihood activity of HT1 households was tea planting, while the main livelihood activities of HT2 and HT3 households were tea planting and non-agricultural employment (Table 4). Similarly, the scale of tea planting for HT1 households was generally larger than that of HT2 and HT3 households in the FWTCS. At present, the price of Fuding white tea is relatively high, but its industry chain is relatively short. Thus, local farmers mainly engage in growing Fuding white tea and directly sell fresh tea leaves to obtain tea industry income, reducing farmers’ livelihood activities.

### 4.2. Income and Welfare of Different Livelihood Strategies

The levels of farmers’ income in the ATTCS were lower than those in the FWTCS, but the proportion of tea industry income to household income was relatively similar between the ATTCS and FWTCS. In the ATTCS, the average level of per capita household income was CNY 15,400, and the average level of household income was CNY 90,400. The average level of tea industry income was CNY 59,400, accounting for 65.71% of the household income. In the FWTCS, the average level of per capita household income was CNY 33,900, and the average level of household income was CNY 173,900. The average level of tea industry income was CNY 113,200, accounting for 65.09% of the total (Table 5).

Based on related research [55] and combined with the characteristics of farmers’ production and living conditions in the tea AHS sites, the indicators of farmers’ welfare consist of the income level of planters in their social circle, satisfaction with the scale of tea gardens operations, and satisfaction with tea planting benefits. The average level of farmers’ welfare in the ATTCS was lower than that in the FWTCS, which is consistent with the relative farmers’ incomes. The income level of planters in their social circle and satisfaction with the scale of tea garden operations in the ATTCS were slightly higher than those in the FWTCS, but satisfaction with tea planting benefits was lower than that in the FWTCS (Table 5).

Compared with agricultural and non-agricultural part-time households, the levels of pure agricultural households regarding income and welfare in the ATTCS were lowest, whereas they were the highest in the FWTCS. In the ATTCS, the average level of HT1 households’ income and welfare were lower than that of HT2 and HT3 households. By contrast, the average level of household income and total welfare of households with different livelihood strategies were ranked as HT1, HT2, and HT3 in the FWTCS. For both, the proportion of tea industry income for HT1 household income was more than 90% (Table 5). Owing to the differences in the relative benefits of agriculture and non-agriculture, the comparative results for the income and welfare of households with different livelihood strategies differ.

After years of rapid development and the continuous expansion of tea production, the market for Anxi Tieguanyin has reached near saturation, leading to stabilized market prices. Hence, farmers in the ATTCS have experienced lower planting benefits and satisfaction. Conversely, the recent rapid growth in the market conditions and size of Fuding white tea has increased farmers’ income, thereby increasing their satisfaction with the benefits of tea planting.

### 4.3. Livelihood Capital of Different Livelihood Strategies

Based on the evaluation indicators of livelihood capital and the data normalization method used in this study, the maximum and minimum values of total livelihood capital of were equal to six and zero, respectively. The maximum and minimum values for each livelihood capital were equal to one and zero, respectively.

Farmer livelihood capital in the ATTCS was higher than that in the FWTCS. The average farmer livelihood capital values in the ATTCS and FWTCS were 1.756 and 1.496, respectively (Table 6). This result is mainly because the ATTCS was protected earlier, and more policies and measures to ensure farmers’ livelihoods have been implemented, such as the insurance system and the subsidy policy for tea planting [56]. However, both values were less than six, indicating that a shortage of livelihood capital. ATTCS households have advantages in terms of natural, physical, human, and cultural capital, while FWTCS households have advantages in terms of financial and social capital. To meet the capital needs of farmers and expand tea production, the FWTCS government launched a credit loan for white tea production. In addition, local governments, associations, and cooperatives of the FWTCS have attached great importance to the development of the tea industry there in recent years. Therefore, farmers have received more social support to engage in tea production.

Compared with agricultural and non-agricultural part-time households, pure agricultural household livelihood capital was not the lowest in the ATTCS and FWTCS. In the ATTCS, the average values of HT1, HT2, and HT3 households’ livelihood capital were 1.739, 1.857, and 1.618, respectively. Similarly, in the FWTCS, the average values of HT1, HT2, and HT3 households’ livelihood capital were 1.454, 1.620, and 1.422, respectively (see Table 6). For both the ATTCS and FWTCS, the livelihood capital of HT1 households was only lower than that of HT2 households, which is inconsistent with many existing studies [48,51]. This result is mainly because the development of the tea industry has provided farmers with more sources of livelihood and directions for labor employment in the tea AHS sites.

In the ATTCS and FWTCS, the average values of households’ cultural capital were the highest, whereas the average values of households’ financial capital were the lowest. In the ATTCS and FWTCS, the average values of household cultural capital were 0.560 and 0.496, respectively, indicating that their cultural capitals were relatively abundant (Table 6). This result is mainly because local farmers have better recognition and cognition of the national culture and traditional customs and can understand the folk tradition and customs, village regulations and agreements, and traditional farming knowledge. However, in the ATTCS and FWTCS, the average values of household financial capital were 0.090 and 0.127, respectively (Table 6). The financial capital of local farmers has been limited by the poor development of the rural financial market. When unexpected risks arise, there is a scarcity of materials to temporarily cope with the risks and provide liquid cash for local farmers.

### 4.4. Effects of Livelihood Capitals on Livelihood Strategies Transformation

Given that the ATTCS and FWTCS are located in Fujian Province, the socio-economic environment faced by the transformation of farmers’ livelihood strategies is relatively similar. Therefore, the livelihood capital indicators selected for the independent variables are the same in this study. For a long time, engaging in agricultural production has been the main livelihood activity of farmers in the ATTCS and FWTCS. Therefore, this study used HT1 households as a benchmark group and explored the significant livelihood capitals indicators for the transformation of households’ livelihood strategies from HT1 to HT2 and HT3.

Based on existing related studies [47] and combined with the data of this study, the estimation model was finally obtained after the multicollinearity test, correlation test, and model adjustment using the Stata MP17 statistical software package. Table 7 presents the effects of livelihood capital on the transformation of farmers’ livelihood strategies.

First, for both the ATTCS and FWTCS, the cultivated tea garden area (NC1) had a significant adverse effect on the transformation of farmers’ livelihood strategies from HT1 to HT2. This result is mainly because the larger the scale of the farmer’s tea gardens, the greater the importance of tea income for them. Consequently, their agricultural production is unlikely to decrease.

Second, in the ATTCS, the degree of dispersion of cultivated tea gardens (NC3) had a significant positive impact on the transformation of farmers’ livelihood strategies from HT1 to HT3. This result is mainly because as the plots of farmers’ tea gardens become more concentrated, they become more conducive to promoting large-scale production, which can reduce labor intensity and save on working time. As a result, they are better able to engage in non-farm work to increase their income.

Third, in the ATTCS and FWTCS, the number of family laborers (HC1) positively affected the transformation of farmers’ livelihood strategies. In the ATTCS, the number of family laborers had a significant positive impact on the transformation from HT1 to HT2. In the FWTCS, the number of family laborers had a significant effect impact on the transformation from HT1 to HT2 and HT3. This result is mainly because the larger the number of household laborers, the more likely they are to engage in non-agricultural livelihood activities. Consequently, rural household livelihood strategies are more likely to be transformed.

Fourth, in the FWTCS, farmers serving as a village cadre (SC3) had a significantly positive impact on the transformation of their livelihood strategies from HT1 to HT2. On the one hand, farmers serving as a village cadre means that their time and energy for engaging in agricultural production activities will be reduced, and at the same time, strengthening tea production is also one of their work tasks. On the other hand, farmers serving as village cadres can accumulate more social resources and receive more social support, making engaging in non-agricultural production activities easier.

Fifth, in the ATTCS, the number of tea planting years (CC3) had a significant positive impact on the transformation of farmers’ livelihood strategies from HT1 to HT2 and HT3. In other words, an increase in farmers’ tea planting years is conducive to promoting part-time occupations in the ATTCS. This result is mainly because farmers gradually become familiar with the tea industry with an increase in tea production time. They often extend the tea industry chain and engage in tea culture tourism reception, tea handicraft production and sales, and other non-agricultural family business types that can promote the diversification of their livelihood activities.

### 4.5. Tea Production and Food Security

Tea leaves are the most important products of the ATTCS and FWTCS. Through compound cultivation in local tea gardens, particularly with “tea and arbor (fruit trees, bamboo, etc.) and tea and crop and herb (grains, tubers, vegetables, and edible fungi, etc.)” systems, both sites produce not only high-quality tea leaves but also a variety of agricultural, forestry, fruit, animal, and aquatic products (Table 8).

The high commodity rate of tea leaves forms a major economic foundation for ensuring food security for local people at the ATTCS and FWTCS sites. In 2022, the per capita tea production was 98.66 kg/person in Anxi County and 104 kg/person in Fuding City (Table 8). According to field investigations, per capita tea consumption was 3.70 kg/person at the ATTCS site and 3.55 kg/person at the FWTCS site. Therefore, most tea that farmers produce was sold as a commodity, providing an essential source of income for achieving food security.

Additionally, the diversified production system ensures that the ATTCS and FWTCS provide local people with abundant and varied food and nutrients, thereby enhancing food security in these regions. The compound cultivation system in tea gardens offers additional food sources for local consumption, particularly grain crops. Although Anxi County and Fuding City are primarily grain-consuming areas in China, their levels of per capita grain production are not very low. In 2022, per capita grain production was 111.85 kg/person in Anxi County and 164.33 kg/person in Fuding City (Table 8). In this way, tea production contributes to promoting local food security.

## 5. Discussion

In terms of livelihood security, farmer income and welfare in the ATTCS were lower than in the FWTCS, but the livelihood capital in the ATTCS was higher than in the FWTCS, mainly because the ATTCS features higher level AHSs and more policies and measures to ensure farmers’ livelihoods have been implemented. Moreover, it is observed here that the levels of livelihood capital for pure agricultural households present an inconsistent relationship with their income and welfare attained across two AHS sites. Factors such as the cultivated tea garden area and the number of family laborers significantly affect farmers’ livelihood strategies, influencing the transformation from pure agricultural to agricultural part-time households at the two tea AHS sites. Furthermore, the high commodity rate of tea products and compound cultivation in tea gardens provide local people with essential sources of income, food, and nutrients, ultimately enhancing food security in these tea AHS sites.

Generally, farmers’ total livelihood capitals at AHS sites are inadequate, with cultural capital being the highest and financial capital being the lowest. In addition, the total livelihood capitals of pure agricultural households are often the lowest [51,57]. In contrast, this study has found that pure agricultural households’ livelihood capitals at both sites are not the lowest, possibly as the result of more local employment opportunities created by the development of the tea industry there. Comparative analyses show that farmers’ livelihood capitals are better in GIAHS than in China-NIAHS, although their income does not necessarily follow this trend, possibly due to different economic development levels [48]. Previous studies have shown inconsistent results when comparing farmers’ incomes with different livelihood strategies across different AHS sites. For example, pure agricultural household income is not necessarily the lowest when compared with agricultural and non-agricultural part-time households [47], which is also supported by this study.

However, this study has the following limitations. First, this study used cross-sectional data for livelihood and food security from the ATTCS and FWTCS. Owing to the lack of baseline data, we could not evaluate the dynamic changes in livelihood and food security. Therefore, the impact of AHS conservation on livelihood and food security requires further exploration. Second, there is a time gap between the data on the livelihood security of farmers in the ATTCS and FWTCS, which affects the results of the comparative analysis to a certain extent.

## 6. Conclusions and Policy Implications

AHSs are the outstanding example of SA, and ensuring livelihood and food security plays a vital role in their conservation and development. This study has evaluated farmers’ livelihoods and food security in the ATTCS and FWTCS and a comparative analysis has been conducted. In addition, a multinomial logistic regression model was used to analyze the impact of farmers’ livelihood capital on the transformation of their livelihood strategies in the ATTCS and FWTCS. The main findings are as follows. First, the level of farmers’ welfare in the FWTCS is higher than that in the ATTCS. Compared with agricultural and non-agricultural part-time households, pure agricultural households’ welfare levels were the lowest in the ATTCS, while their welfare levels were the highest in the FWTCS. Second, the area of tea gardens and the number of laborers were factors that significantly affected the transformation of farmers’ livelihood strategies from pure agricultural households to agricultural part-time households in the ATTCS and FWTCS. Third, the high commodity rate of tea products, combined with the compound cultivation in tea gardens, provides local people with essential sources of income, food, and nutrients, so as to improve food security in the ATTCS and FWTCS.

This study’s findings have important policy implications. First, the governments of tea AHS sites should accelerate the development of tea deep-processing products, especially tea-related healthcare products, to further explore and utilize tea’s food and nutritional value. Second, measures for the livelihood security of farmers at the AHS sites should be strengthened. Farmer livelihood capital should be improved from many aspects, such as the improvement of rural financial services. Conservation efforts should prioritize increasing farmers’ income and improving their living standards and welfare. Third, to further enhance the sustainable development of agricultural systems, SA should integrate the wisdom, experience, and lessons from AHSs. Since different types of AHSs encompass a wide range of sustainable agricultural practices, SA can be more effectively applied and adapted to diverse areas by leveraging the strengths of various AHSs.

## Figures and Tables

**Figure 1 foods-13-02238-f001:**
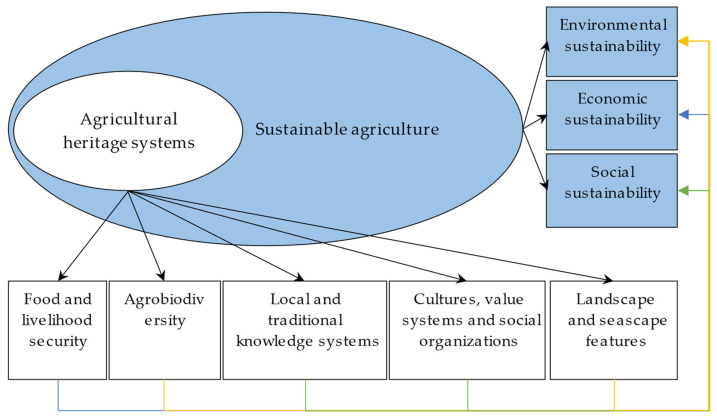
The relationship between agricultural heritage systems and sustainable agriculture.

**Figure 2 foods-13-02238-f002:**
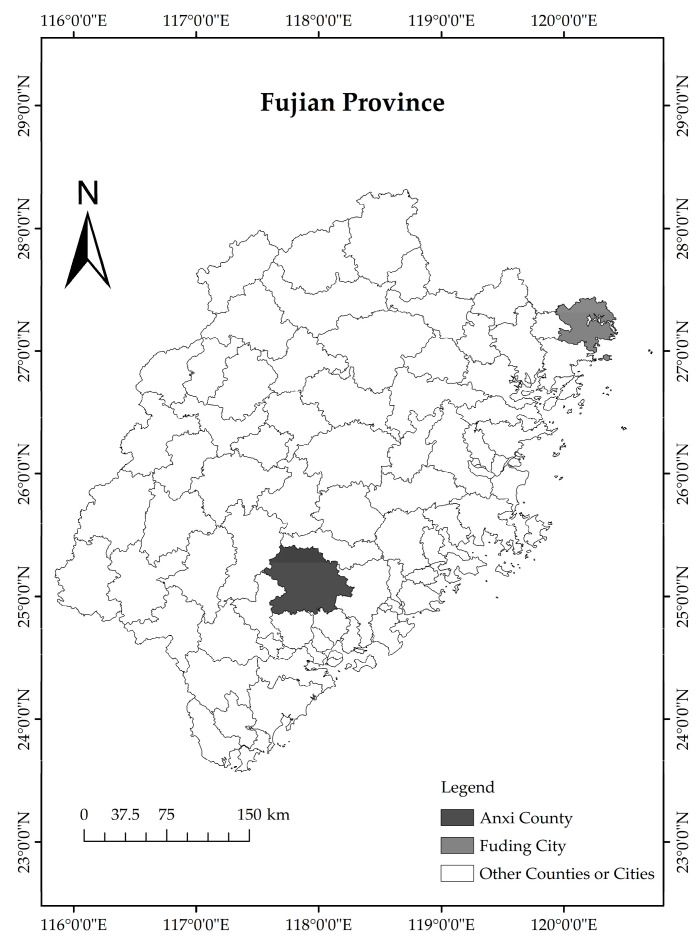
The locations of the ATTCS and FWTCS.

**Figure 3 foods-13-02238-f003:**
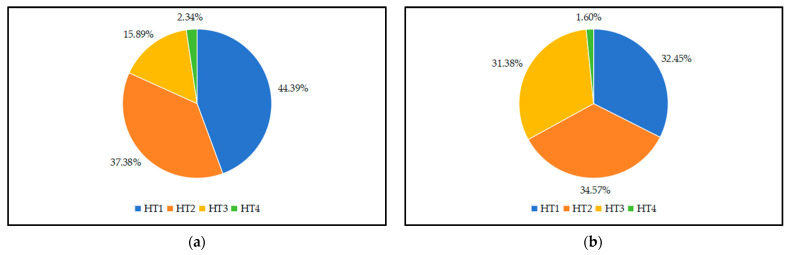
The proportion of households with different livelihood strategies. (**a**) ATTCS; (**b**) FWTCS.

**Table 1 foods-13-02238-t001:** The number of surveyed households in the ATTCS and FWTCS.

AHS	Town	Surveyed Villages	Surveyed Households
ATTCS	Lutian	3	42
	Xiping	8	131
	Huqiu	3	41
FWTCS	Guanyang	3	35
	Diantou	3	38
	Bailin	4	43
	Panxi	4	78
	Taimushan	2	23
Total		30	431

**Table 2 foods-13-02238-t002:** Demographic characteristics of households in the ATTCS and FWTCS.

		ATTCS	FWTCS
Item	Category	Percent (%)	Percent (%)
Gender	Male	92.06	84.79
	Female	7.94	15.21
Age (years old)	≤35	5.61	8.76
	36~45	16.82	18.43
	46~55	30.37	35.48
	56~65	31.78	25.81
	>65	15.42	11.52
Education	Below primary school	3.27	22.12
	Primary school	22.90	19.82
	Junior high school	53.27	30.88
	Senior high school	17.76	16.59
	University or above	2.80	10.60
Household size (persons)	1~3	10.75	15.67
	4~6	58.41	64.98
	7~9	17.29	15.67
	≥10	13.55	3.69
Tea gardens area (hectares)	≤0.13	21.96	5.53
	0.13~0.40	43.46	35.02
	0.40~0.67	18.22	32.72
	0.67~0.93	7.94	6.91
	>0.93	8.41	19.82

**Table 3 foods-13-02238-t003:** Indicators of livelihood capital.

Type	No.	Indicator Name	Indicator Definition
Natural capital *	NC1	Cultivated tea garden area	Actual quantity
	NC2	Quality of cultivated tea gardens	1 = Very poor, 2 = poor, 3 = average, 4 = better, 5 = very good
	NC3	Dispersion degree of cultivated tea gardens	1 = Very dispersive, 2 = relatively dispersive, 3 = average, 4 = relatively concentrated, 5 = very concentrated
	NC4	Distance between residence and cultivated tea gardens	1 = Very far, 2 = relatively far, 3 = average, 4 = relatively near, 5 = very near
Physical capital	PC1	Structure of current housing	1 = Adobe, 2 = brick–wood, 3 = brick–concrete, 4 = building
	PC2	Quantity of durable consumer goods	Actual quantity, including air conditioning, motorbikes, mobile phones, and much more
	PC3	Conditions of agricultural production tools	1 point = draught animal, 2 points = hand farm implement, 3 = small and medium-sized agricultural machinery, 4 = large-sized agricultural machinery
Financial capital	FC1	Conditions of household savings	1 = None, 2 = below CNY 50,000, 3 = CNY 50,000~100,000, 4 = CNY 100,000~150,000, 5 = over CNY 150,000
	FC2	Conditions of household debts	1 = None, 2 = below CNY 50,000, 3 = CNY 50,000~100,000, 4 = CNY 100,000~150,000, 5 = over CNY 150,000
	FC3	Conditions of livestock and poultry	Converted to pig equivalents, where 1 cow = 5 pigs, 1 sheep = 3 pigs, 30 chickens or ducks = 1 pig, 15 geese = 1 pig
Human capital	HC1	Number of family laborers	Actual quantity
	HC2	Education of the head of the household	1 = Below primary school, 2 = primary school, 3 = junior high school, 4 = senior high school, 5 = university or above
	HC3	Health of the head of the household	1 = Very poor, 2 = poor, 3 = average, 4 = better, 5 = very good
	HC4	Times of agricultural skills training	Actual quantity
Social capital	SC1	Degree of neighborhood communication	1 = Very little, 2 = relatively little, 3 = average, 4 = more, 5 = a lot
	SC2	Joined a cooperative or association	0 = No, 1 = Yes
	SC3	Serve as a village cadre *	0 = No, 1 = Yes
Cultural capital	CC1	Understanding of traditional agricultural technique	1 = Worst, 2 = bad, 3 = average, 4 = well, 5 = best
	CC2	Understanding of traditional agricultural culture	1 = Worst, 2 = bad, 3 = average, 4 = well, 5 = best
	CC3	The number of tea planting years	Actual quantity

* In the ATTCS and FWTCS, tea gardens are the most critical land resource for farmers. Therefore, the operation of tea gardens is used to represent farmers’ land management. The village cadre refers to grassroots government workers responsible for managing and organizing villagers in rural China.

**Table 4 foods-13-02238-t004:** The livelihood activities of households with different livelihood strategies in the ATTCS and FWTCS (Unit: %).

	ATTCS				FWTCS			
Livelihood Activities	HT1	HT2	HT3	Average	HT1	HT2	HT3	Average
Tea planting	98.95	100	100	99.53	100	100	100	98.94
Other crops planting (Excluding Tea)	45.26	35.00	26.47	37.85	27.87	41.54	27.12	31.91
Livestock and poultry breeding	89.47	92.50	91.18	91.12	8.20	12.31	8.47	9.57
Non-agricultural employment	6.32	100	100	58.41	34.43	100	100	78.72

Note: Analysis of HT4 households was not conducted because of the small sample size.

**Table 5 foods-13-02238-t005:** The income and welfare of different types of households in the ATTCS and FWTCS.

AHS	Livelihood Outcomes	Indicators	HT1	HT2	HT3	Average
ATTCS	Farmers’ income	Household income (10^4^ CNY)	8.26	8.80	12.04	9.04
		Tea industry income (10^4^ CNY)	7.59	5.55	3.06	5.94
		Per capita household income (10^4^ CNY)	1.45	1.53	1.85	1.54
	Farmers’ welfare	Income level of planters in their social circle	2.67	2.95	2.82	2.80
		Satisfaction with the scale of the tea gardens operations	0.79	0.73	0.62	0.74
		Satisfaction with tea planting benefits	3.16	3.11	3.24	3.15
		Total	6.62	6.79	6.68	6.69
FWTCS	Farmers’ income	Household income (10^4^ CNY)	18.29	17.74	16.51	17.39
		Tea industry income (10^4^ CNY)	17.51	11.77	4.96	11.32
		Per capita household income (10^4^ CNY)	4.02	3.23	2.97	3.39
	Farmers’ welfare	Income level of planters in their social circle	2.82	2.78	2.54	2.71
		Satisfaction with the scale of the tea gardens operations	0.70	0.62	0.37	0.56
		Satisfaction with tea planting benefits	3.69	3.78	3.10	3.51
		Total	7.21	7.18	6.02	6.78

**Table 6 foods-13-02238-t006:** Livelihood capitals of different types of households in the ATTCS and FWTCS.

	ATTCS				FWTCS			
Livelihood Capitals	HT1	HT2	HT3	Average	HT1	HT2	HT3	Average
Natural capital	0.225	0.236	0.208	0.227	0.185	0.192	0.172	0.182
Physical capital	0.293	0.322	0.263	0.296	0.139	0.144	0.125	0.136
Financial capital	0.087	0.090	0.093	0.090	0.124	0.149	0.111	0.127
Human capital	0.286	0.298	0.240	0.280	0.203	0.235	0.218	0.220
Social capital	0.296	0.350	0.237	0.303	0.291	0.390	0.328	0.335
Cultural capital	0.552	0.562	0.576	0.560	0.512	0.512	0.468	0.496
Total	1.739	1.857	1.618	1.756	1.454	1.620	1.422	1.496

**Table 7 foods-13-02238-t007:** The model results of effects of livelihood capitals on livelihood strategies transformation.

		ATTCS				FWTCS			
		HT2		HT3		HT2		HT3	
		Coefficient	P >|z|	Coefficient	P >|z|	Coefficient	P >|z|	Coefficient	P >|z|
Natural capital	NC1	−0.119 **	0.032	−0.008	0.322	−0.102 ***	0.003	−0.000	0.942
	NC3	0.169	0.353	0.287 **	0.033	0.022	0.630	−0.053	0.302
	NC4	−0.098	0.595	−0.075	0.575	0.031	0.837	0.151	0.292
Physical capital	PC1	0.432	0.167	0.098	0.662	0.278	0.317	−0.186	0.472
Financial capital	FC2	0.212	0.379	0.136	0.452	−0.125	0.333	−0.087	0.466
	FC3	−0.071	0.406	−0.016	0.380	0.601	0.322	0.693	0.204
Human capital	HC1	0.189 **	0.023	0.091	0.190	0.465 ***	0.005	0.273 *	0.053
Social capital	SC3	0.395	0.448	0.481	0.195	0.857 *	0.055	0.646	0.139
Cultural capital	CC3	0.065 ***	0.002	0.035 **	0.023	−0.022	0.195	−0.016	0.299
Constant		−5.189 ***	0.002	−2.770 **	0.014	−0.242	0.294	−0.108	0.923

Notes: HT1 is a benchmark group; *** 1% significance, ** 5% significance, and * 10% significance.

**Table 8 foods-13-02238-t008:** The output of major agricultural products of Anxi County and Fuding City in 2022.

	Anxi County		Fuding City	
Agricultural Products	Yield (10^4^ Tons)	Per Capita Production (kg/Person)	Yield (10^4^ Tons)	Per Capita Production (kg/Person)
Tea	8.27	98.66	4.09	104.00
Grain	9.38	111.85	6.46	164.33
Vegetable	27.47	327.59	19.14	486.46
Fruit	1.74	20.79	4.40	111.94
Livestock and Poultry	3.08	36.71	0.75	18.95
Aquatic Products	0.16	1.95	1.11	28.21
Edible Fungi	0.76	9.11	1.80	45.82

Notes: Available online: http://fjax.gov.cn/zwgk/zfxxgkzl/zfxxgkml/tjxx/202310/t20231010_2949847.htm, http://www.fuding.gov.cn/zwgk/tjxx_0808/tjgb/202305/t20230508_1756251.htm (accessed on 10 May 2024).

## Data Availability

The original contributions presented in the study are included in the article, further inquiries can be directed to the corresponding author.

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
