# Peer review of "Livelihood and Food Security in the Context of Sustainable Agriculture: Evidence from Tea Agricultural Heritage Systems in China"

_foods, 2024, doi:10.3390/foods13142238_

Round 1

Reviewer 1 Report

Comments and Suggestions for Authors

First, I would like to thank you for the opportunity to review the article entitled ,, Sustainable agriculture and food and livelihood security. Evidence from Tea agricultural heritage systems in China,,. The article is interesting, and the subject of the paper fits the field of interest of Foods.

Dear authors, in the introduction section lines 74-91,you mention the objectives of the present paper, such as crop systems, farmers classification, strategies and living activities, as well as the effects of living capitals, but do not specify how are the beneficiaries of this research paper. Also, how does your aim and objectives if answered reduce the gap in academia?

I strongly advise to mention these aspects as well.

The materials and method section looks good.

The results section. Try to discuss results with recent literature and provide reasoning of responses recorded. Improve the discussion with logical and reasoning approaches.

The authors do not discuss possible external validity of the results in terms of possible insights in other countries/regions as well additional aspects that they would have liked to better answer to their research question.

I hope that these comments will help you with your work.

Best regards.

Reviewer 2 Report

Comments and Suggestions for Authors

1.       In the introduction section, the authors refer to SDG2. While many readers may be familiar with the various SDGs, others may not. The authors need to explain SDGs. This could be done as a footnote if necessary.

2.       I am not sure that providing the latitude and longitude coordinates of the study areas (page 6) is necessary. If the authors want to provide location information, a map showing the locations of the two agricultural systems under study would have more utility.

3.       The size of the pie charts (Figure 1) need to be increased. In their present form, they are difficult to read.

4.       On page 12, the authors refer to “village cadres”. The authors need to explain the meaning of this term.

Comments on the Quality of English Language

While the quality of the written English is good, the paper does need proofreading by a native English speaker.
